# Health Professionals’ Experiences with Health-Promoting Dialogues for Older Home-Dwellers—A Qualitative Study

**DOI:** 10.3390/bs14060464

**Published:** 2024-05-30

**Authors:** Liza Wigaard Johansen, Hilde Lausund, Nina Jøranson

**Affiliations:** Faculty of Health Sciences, VID Specialized University, 0319 Oslo, Norway; liza.w.johansen@gmail.com (L.W.J.); hilde.lausund@vid.no (H.L.)

**Keywords:** health-promoting dialogues, preventive home visiting, home-dwellers, empowerment, focus groups, thematic analysis

## Abstract

Home-dwelling older people without healthcare services might develop vulnerability and health-related issues that should be detected proactively by service providers. Health-promoting measures directed towards the target group could facilitate living longer and better at home, as well as delay the need for healthcare services. One approach is through health-promoting dialogues between the municipality and healthcare professionals. This study aims to explore the experiences of healthcare professionals involved in health-promoting dialogues with home-dwellers aged over 75 years without health service decisions in Norway. Data were collected through three focus groups. Thematic analysis was applied to the data resulting in the emergence of one major theme, “challenging dialogues”, comprising three sub-themes: “promote the individual’s perspectives”, “uncovering vulnerability”, and “ambiguity of the dialogues”. The health-promoting dialogue uses a resource perspective for the elderly to remain independent in old age and can reveal vulnerability and underlying needs. The purpose of the dialogue appears ambiguous for the target group, which leads to unclear service expectations and frequent rejections of the offer. Nevertheless, this health-promoting service has a clear purpose of identifying and meeting the needs of the target group in a broader sense during the ageing process.

## 1. Introduction

Forthcoming Norwegian generations are projected to live longer and face old age with better finances, higher education, better health, and completely different material conditions compared to today’s older generation [1]. Old age leads to increasing vulnerability and risk for diseases, functional decline, and multimorbidity, causing more complex care needs and health challenges in daily life [2]. The risk of developing frailty increases with age, and, according to the Norwegian Directorate of Health [3], this includes almost 25% of all people over 85. This development supports the significance of health by promoting an early focus on older people, aiming for healthy ageing as far as possible for the target group [1].

Population projections indicate that Norway’s population over 70 years old is expected to almost double by 2060 [4]. The 67–80 age group is expanding the most, while older people over 80 are expected to have greater growth closer to 2030, affecting the demand for healthcare services [5]. This development underpins the prioritisation of early interventions, particularly before the age of 80, to prevent and delay the need for healthcare services as long as possible. Early intervention in this target group is in line with policy recommendations underscoring the use of one’s own resources, abilities, and opportunities to master everyday life [5]. Increased investment must be made in resource-oriented measures to prevent health issues and promote quality of life [5,6]. In Norway, there has been an increasing emphasis on health-promoting measures and empowerment due to changes in health policy in the last decades [5,7,8]. This development supports investment in preventive and health-promoting services at home. In this regard, the reinforcement of preventive measures, such as preventive home visits to older people (PHV) in all municipalities, is considered an important intervention and was facilitated for the municipalities through guidance materials provided by the Norwegian Directorate of Health in 2017 [9]. Most studies on PHV are based on the recipients, with a qualitative design describing their experiences, showing positive perceptions of the measure, increased feelings of safety, useful information, and valuable conversations [10,11,12].

The measure of *health dialogue*, also called *health-promoting dialogue*, stems from PHV to prevent illness and injury as well as support health, functional capacity, and participation [13]. Health-promoting dialogue is an outreach and proactive measure initiated by the municipality aimed at older people (75+) living at home who do not seek or receive healthcare services [10]. The design of such publicly financed dialogues varies between municipalities but often lasts for one hour and is usually provided once, although it could be repeated based on request or if necessary [12]. A facilitated guide describing health-related topics containing information on activity options and exercise offers, along with leaflets comprising relevant information about municipal services, is often used to support consistency in the dialogue. In the Norwegian context, the dialogue often focuses on resources, seeking to stimulate reflections on how older people could handle and plan their own health and life situations to remain independently at home in the future [14,15]. The dialogue provides tailored guidance on health-promoting lifestyles as well as information on the municipalities’ various relevant offers and services [13]. The health-promoting perspective [16] is expressed by focusing on the individual’s resources and self-mastery capability and motivating older people to take proactive steps to maintain their health and well-being. A subjective understanding of health through health-promoting dialogues taking a resource-oriented approach, as highlighted in this paper, renders the perception of health as a vital component for a good life, positive functioning, and coping, which can be promoted in daily life [17].

Denmark has been a pioneering country in PHV, legislating the annual offer in 1996 at the age of 75, and Australia has a legalised offer [18]. Other countries have gained significant experience with PHV; however, studies describe great variations regarding implementation, focus, content, and effects across the globe [13,19,20]. The provided measures are therefore difficult to compare [21], and this is also true in the Norwegian context [13]. In Norwegian municipalities, PHV has developed from individual mapping of possible functional impairment, need for help, and prevention of injuries towards the development of health-promoting dialogues focusing on resources and self-management, as well as health counselling and service information [12].

PHV is conducted by professionals, and nurses are the most frequently professionals to carry out PHV [22]. There are, however, few studies investigating healthcare professionals’ perspectives on PHV [13]. One Norwegian study investigating nurses’ experiences with PHV in one municipality describes how structural, professional, and contextual factors influence nurses’ ability to support the recipients’ changing needs and independence [23]. A Swedish study on the imposed offer of PHV at the age of 75 found that nurses struggle with balancing their personal vs. professional approach, orientation, and focus on resources vs. perspective of illness [24].

The World Health Organisation (WHO) recognises empowerment as a key concept in health promotion [25]. The term *empowerment* refers to a positive view of people’s resources and their capacity to act and “make a difference” [26]. As an actively acting subject, individuals can do their best under given conditions [27]. Empowerment can be understood as a process where individuals and groups are encouraged to mobilise power to gain increased self-confidence, enhanced self-image, increased knowledge and skills, and more power, management, and control over one’s own life. Such a process can enable individuals to realise their own resources, coping strategies, and opportunities to initiate various forms of lifestyle change [28,29]. Empowerment processes can be considered a framework for health promotion [30], indicating that individuals take responsibility for their own health, get acknowledged for their personal expertise, and actively participate in caring for their own health. An empowerment process is thus the conceptualisation of individuals to optimise health and meaningfulness in life [31], which has also been described as a “coming from within” approach directing individual solution and growth [32]. In this regard, the emphasis on the empowerment process in the health-promoting dialogues can affect the perception of older people towards their own power and behaviour, thereby influencing their control over their own health behaviours.

There is a need for municipalities to share experiences and a need for knowledge of models for executing PHV and health-promoting dialogues. There is also a need for additional knowledge regarding the academic competence of healthcare professionals conducting meetings with older home-dwellers under such measures [7,13]. This study aims to explore healthcare professionals’ experiences with health-promoting dialogues with older home-dwellers over 75 years old and the potential of such dialogues.

## 2. Methods

### 2.1. Study Design

The study embraces an explorative design with qualitative methods, and data were collected through focus group interviews. Focus groups provide good insights into themes lacking knowledge [33]. The methodological strength lies in using group dynamics as a means of producing more complex data through the joint exchange of opinions, which will highlight different understandings and attitudes. Group dynamics promote the participants’ exploration and clarification of each other’s perspectives and create reflection on their own opinions. Such production of empirical knowledge through social interaction cannot be achieved through individual interviews [34].

### 2.2. Sample and Data Collection

Participants were recruited through direct contact with four existing measures from four municipal services providing health promotion dialogues. Potential participants indicated their interest through the municipality, and contact was made through snowball recruitment. The sample consisted of healthcare professionals with different professions and a minimum of 2 years of experience with preventive and health-promoting approaches towards older people, preferably related to health-promoting outreach approaches towards older people without health service decisions. A total of nine female participants from four different municipalities providing such dialogues produced a convenience sample, where all participants had various long-term professional experiences gained from numerous realms of healthcare services. Two participants were nurses and seven were occupational therapists, and all had experience with professional development and project work related to health-promoting conversations. Most of them had long work experiences in various realms of the healthcare service, such as mental health work and geriatrics, rehabilitation, public health, and health supervision. The sample was divided into three focus groups based on three professionals who contacted potential participants from their own network. This snowball recruitment [35] therefore resulted in three different groups. The focus group interviews were conducted by the first author in March 2021 and lasted for one hour. The article is based on a Master’s thesis previously submitted at VID Specialised University in Oslo, Norway [36], which explores a relatively unknown topic within the field of health promotion.

Due to the COVID-19 pandemic, the focus group interviews were conducted via digital platforms. The different focus groups consisted of participants from the same workplace with similar backgrounds in different parts of Norway. The informants in each focus group largely knew each other prior to the interview, which facilitated appropriate group dynamics [35]. However, during the interviews, the participants commented on and bolstered each other’s experiences and perceptions, which made the group dynamics seem flexible and spontaneous.

The following research questions guided the study: “What experiences do healthcare professionals have with health-promoting dialogues and their role in these conversations? What benefits and developmental opportunities can such a conversation bring?” A semi-structured interview guide was used to collect data, and three topics were discussed, such as their approach to the health-promoting dialogue, including their opinions on health promotion, the competence of healthcare professionals conducting the conversations, and organisation and development opportunities for such conversations.

### 2.3. Analysis of Data

The three focus group interviews were recorded and transcribed verbatim by the first author. The data were analysed through thematic analysis (TA), which is suitable for describing and identifying one or more themes. Our approach was inspired by the six-step model described by Braun and Clarke [37] in order to identify and report patterns through themes derived from the empirical material related to the research questions [37,38]. The analytical exploration starts with a thorough reading of the material, which informs the early phase of the analysis in search of meaningful content. Subsequently, preliminary codes were proposed based on interesting features identified through an inductive approach to the data material. The next step was the development of preliminary sub-themes and themes based on the previous steps of the analysis and on the research questions. Such analysis of qualitative data is an interpretative and reflexive process in which understandings emerge by moving back and forth between the analysis steps, facilitating engagement with the analysis process along with deep reflections and discussions. Themes and sub-themes developed further through writing the report before the findings were projected. The analysis was mainly performed by the first author and supervised by the last author; thus, both authors met several times to converge and agree on the emerging themes. Finally, all authors agreed on the final themes.

It is remarkably significant to identify the meaning behind any theme, and the themes should be given clear and identifiable distinctions [37,38]. The relationship between codes, themes, and different levels of themes and sub-themes was thoroughly assessed, interpreted, and adjusted through repeated and mixed movements between the different steps of reflexive processes described by Braun and Clarke [39] as reflexive TA. Themes do not, however, emerge from data but appear as “interpretative stories about the data”. TA is thus a reflexive process rather than a step-by-step procedure for analysing data.

### 2.4. Ethical Considerations

All participants received written and oral information about the study prior to their recruitment and were informed that their participation was voluntary and anonymous. Subsequently, the participants provided written informed consent. The study was approved by the Norwegian Agency for Shared Services in Education and Research (project number 614852) and complies with guidelines by the National Committee for Research Ethics in the Social Sciences and the Humanities [40].

## 3. Results

The findings have been organised under one main theme, “Complex Conversations”, which is divided into three related sub-themes: “Promote the individual’s perspectives”, “Uncovering vulnerability”, and “Ambiguity of the dialogues”. An overview of the theme and the sub-themes is presented in Figure 1.

### 3.1. Theme: Complex Conversations

Important notes for healthcare professionals throughout trust-based conversations with the recipients grant access to the individuals’ perspectives by focusing on resources, coping skills, and awareness towards their health. Further, the participants described the significance of the dialogues, eliciting reflections on uncovering vulnerabilities in older people linked to sensitive topics. The analysis also showed how the intention of the health-promoting dialogue could appear unclear to those who received the offer. It may seem that the recipients’ understanding of such a health-promoting measure may be perceived as ambiguous based on unclear information about the offer.

#### 3.1.1. Sub-Theme 1: Promote the Individual’s Perspectives

The health-promoting view in the dialogues was carried out through a facilitated retrieval of individual perspectives and resources from the older person. The participants identified the “person in the situation” by talking about sources of recipients’ meaning and mastery in life, exemplified through metaphors such as “mobilising their power” and “finding the resources, watering them, and making sure they grow and become stronger”. The participants identified the recipients’ personalities, values, and interests, as well as their wants and needs, to suggest relevant and meaningful activities and measures. They experienced the recipients being interested in increasing awareness about health issues, planning for old age, and fulfilling wishes related to retirement.

It was crucial for the participants to be mentally present in their dialogues and focus on the individuals’ wants and needs by being fully engaged and attentive. Acknowledging the person creates meaning and trust through the dialogue. One of the informants reflected on this importance by referring to the well-known Danish philosopher Søren Kierkegaard’s words on the art of helping, which primarily concern the enhancement of the helper’s moral responsibility and the dialogue with the patient [41]:


*“When I worked with rehabilitation, Kierkegaard was of help through the ’art of helping’, which is about the fact that to help someone else, you need to find him where he is, and start from there. I try to keep that in mind.”*
(Participant 2)

The health-promoting dialogues require a trustworthy relationship, particularly in the first meeting with the municipality’s services for many potential recipients. The participants underlined the significance of conducting a successful first meeting, which could increase the chances that those who accepted the offer would contact again. Further, they described the facilitated dialogue guide as a useful starting point for the conversation to revolve around individual health situations and needs. They also expressed the importance of being sensitive to the recipients’ needs during the dialogue, as one participant described:


*“We don’t have to get through everything we’re supposed to. Sometimes it is just about putting everything aside and talking about what they want to talk about.”*
(Participant 6)

The participants showed a clear person-oriented angle in the health-promoting dialogues through active presence and a holistic view of health towards the older recipients. This approach was carried out by asking the question “What is important to you?” to explore the potential needs of the person. They found this question to shed light on older people’s perspectives, plans, and goals for good ageing processes, in addition to focusing on maintaining and optimising well-being, coping, and quality of life. It seems that combining the exploration of what is important for the individuals with the dialogue guide could result in targeted conversations.

#### 3.1.2. Subtheme 2: Uncovering Vulnerability

The health-promoting dialogues also provided an opportunity to uncover the residents’ vulnerability. Opening up to such issues might instil resistance among the recipients, which demands a unique sensibility in the participants. Several participants emphasised that they had acquired increased expertise on topics affecting both physical and mental health in old age, which made them better equipped to recognise vulnerability factors. The participants described the significance of daring and asking challenging questions related to topics that could be perceived as sensitive by the recipients, such as transition phases, grief processes, depression challenges, sexuality, lack of relationships, loneliness, and old age. This underlines the importance of trying to identify the older individual’s underlying needs during the dialogue, as described by one of the participants:


*“For me, a health-promoting conversation is about trying and figuring out where to start. After all, we have these topics from the dialogue guide; then it is a matter of trying and discerning within which of these topics we can have an opportunity to help shine a light on which they can work a bit.”*
(Participant 2)

The health-promoting dialogues were also subject to an assessment of lifestyle changes in older people, which could also result in resistance from the recipients. Such ambivalence led several participants to develop strategies. One participant called this approach “to roll with resistance”, referring to exploring individuals’ attitudes and relationships with their habits and their willingness to alter them to maintain good health and safeguard independence. This approach was highlighted by several participants:


*“For me, it means starting from what that person has inside of them both in terms of personality, hobbies, and interests and finding out what that person has liked to do, likes to do now, and trying to build on that.”*
(Participant 7)

The quote shows the importance of exploring opportunities, needs, and ambivalence associated with one’s health and lifestyle changes.

Further, several participants described how the health-promoting dialogues could decipher attitudes towards one’s ageing process. In addition, the dialogues touched upon a wide range of health-related topics on which there had been greater awareness, such as sleep, alcohol consumption, and networks, in addition to relationships, sexuality, and identity issues.


*“Asking about alcohol habits was not something we did before, as it was too personal. (…) We realised that it was a very important topic to talk about with those we visited, and now we are doing it. (…) We ask just as much about their alcohol habits as how much water and liquids they take in.”*
(Participant 1)

#### 3.1.3. Sub-Theme 3: Ambiguity of the Dialogue

The age-specific limits for the target group were adjusted between the ages of 75 and 80 following the evaluation of feedback given by the recipients of the offer. The age variations are due to the feedback that the offer was perceived as irrelevant to potential recipients. The participants consistently experienced that many recipients felt that this healthcare service comes too early in the age course. They understood the recipients to perceive themselves as able-bodied, thus perceiving the offer as a little irrelevant. The participants believed that older people living with a health condition that is not yet perceived as limiting their quality of life and life expression would, unfortunately, decline the offer happily.


*“That is what is positive about the outreach part of the business that you have the opportunity to approach before it starts to go down (…). I also find that some of the 77-year-olds whom we seek out refuse our offer. And I think that has a lot to do with the fact that you’re not there mentally, that you understand your potential situation in a few years as you start to get older.”*
(Participant 5)

The participants experienced that several older people agreed to the health dialogue because they assumed that the offer would have exclusively disease and injury prevention purposes, thus falling in line with understandings of the traditional preventive home visit (PHV). Their expectations related to wound care and drug-related clarifications were also heightened:


*“My predecessor was a nurse and wore blue plastic covers on the outside of her shoes, and immediately sent some signals that gave a completely different focus: “Can you look at this wound?” There are many people who think we come home to remove carpets, but that’s really the last thing we do.”*
(Participant 4)

They experienced many declines for various reasons, such as the experience of being in physical shape, a lack of information, and unclear expectations regarding the purpose of the dialogues. Several participants believed that such a health-promoting measure should be able to meet needs to a greater extent. The municipalities represented in this study offered health-promoting dialogues to residents over the age of 80, except for one municipality with an age limit of 78 years. At the same time, some participants wished that the offer should cater to even younger age groups than the current target group to a greater extent, indicating that such a measure should also address the broader needs in the ageing process, such as transition phases, for example, in the event of the loss of a partner, or how to help people with dementia.


*“We have something to gain in terms of being able to catch vulnerable older people’s earlier health than when they turn 80. I think that there is a potential that is socially and economically profitable as well.”*
 (Participant 3)

Several participants experienced that older people expressed scepticism and ambivalence towards the offer but altered their perception and found the dialogues to be useful afterwards. The participants described a significant proportion of the target group who did not see the need for such a health-promoting and preventive measure, most likely due to their ignorance regarding the offer’s intention, content, and utility.

## 4. Discussion

Our findings reflect the potential of health-promoting dialogues to increase older people’s awareness of potential health challenges and opportunities for coping. Further, the study findings show that vulnerability can be identified through dialogues with healthcare professionals, which also affect older people’s experiences of their own health and life situations. In addition, the findings show how the participants witnessed many recipients’ perceptions of the offer as ambiguous. These findings raise several challenges in the health-promoting dialogue and will be discussed in light of empowerment, the opportunity for health promotion, and the overall mandate of the health-promoting dialogues.

### 4.1. The Health-Promoting Opportunity Space

The dialogues focused on how healthcare personnel could promote the individual’s perspectives on issues concerning the recipients’ present and future situations. Further, they actualised thoughts about planning one’s old age and wishes related to retirement. This approach complies with a resource focus in meetings with the target group through the facilitated dialogue guide by highlighting a focus on the older people’s coping skills and awareness of their health and is in line with policy documents underpinning independent living when ageing in place [1,5]. The resource focus approach is recognised in how health-promoting healthcare services for older people are supposed to include customised training, advice, support, and guidance on lifestyle and self-management [5,42]. The significance of such a focus is confirmed in the Norwegian study by Dale and Westbye [10] on the target group’s experiences with health-promoting dialogue. They reported that the recipients expressed the security of being confirmed as valuable, and the dialogue allowed them to gain faith in their ability to preserve their own health. This is in line with our findings showing that our participants perceived the dialogues to contribute to increasing the older people’s responsibility for their own health and the significance of the fact that healthcare professionals in such outreach services should stimulate the older people’s awareness and understanding of their situation to gain increased control over action alternatives. Strengthening older home-dwellers’ prerequisites to be able to handle everyday activities in line with their own goals and wishes is part of the overall health-promoting policy towards older people [5,13]. These prerequisites follow individual objectives in health promotion work, which include increasing personal knowledge and mobilising and utilising the resources, abilities, and opportunities the person possesses while coping with everyday life. This approach aligns with the empowerment mindset of individual involvement and user participation [32], which in turn aims to strengthen individuals’ opportunities to take responsibility for improving their own health and life situations [2,17,43].

The purpose of the dialogues was also to establish a mutual and safe relationship, aiming to uncover needs that were put forward by the older people themselves. Several of our informants considered it their task to convey openness about topics that were considered difficult to talk about, for example, physical and mental health, family relationships, social networks, etc. Health-promoting dialogues might provide older home-dwellers with a feeling of not being forgotten [18] and an experience of increased safety because one is noticed early in old age and is provided “an open line” towards the healthcare service when necessary [10]. However, the dialogue might be more useful for people in vulnerable life situations or with reduced social networks and thus provide the opportunity to accommodate psychosocial support [17]. Nevertheless, when a health-promoting dialogue provides the opportunity to establish a personal relationship between an older home-dweller and the municipality’s service, a connection is established that the system would otherwise not have encountered [18].

Our informants considered it crucial to build trust during the dialogue, which became a key prerequisite for discovering a vulnerability in older people with subsequent needs assessments. This central finding has been confirmed in several studies, emphasising the importance of spending time when building trust in the relationship and supporting emotions to create security so that older people dare to share sensitive information about their everyday lives and health issues [10,24,43,44]. A trusting relationship underpins a dialogue about needs and health resources [23,24], although it takes time to develop a safe and trusting relationship. Older people could be less talkative, which might challenge the possibility of building trust in the conversation. At the same time, it may also happen that healthcare professionals find it challenging to ask intimate questions out of fear of humiliating the older person [24]. Despite this issue, the Norwegian study by Dale and Westbye [10] also found that several older people who accepted the offered dialogue also wanted follow-up consultations, which were provided in several municipalities [22]. These studies confirm that repeated conversations with the same healthcare professional over time can develop trustworthy relationships with the municipality.

However, our informants also reported on older home-dwellers who appraised the dialogue offer to be of little use to them because they considered themselves active and had self-management abilities. Further, some potential recipients were found to decline the offer due to an expectation of a disease-oriented focus on the content, for example, wound care and drug management. This finding describes a rather common misunderstanding of the dialogue as the first step towards initiating home care services. The issues of unclear purposes and aims reflect the ambiguity of the health-promoting dialogues, leading to ambiguous focus and unclear expectations among the target group [23]. A Norwegian study investigating self-care ability in older people found the vast majority in the target group (mean age 75 years) possessing a high level of self-care ability [45], which might explain why only half of the target group accepts the health-promoting offer [13]. They are likely not considering themselves as the target audience for the intervention. Behm et al. [46] confirmed that scepticism towards the content of the dialogues may be due to a lack of familiarity with the purpose of the health-promoting perspective, which in turn will influence the understanding of the dialogue. When potential recipients believe that the dialogue focuses on disease prevention and not health promotion, it might result in a lack of attention to questions related to personal health or future situations, which in turn might prevent a satisfactory benefit from the offer [23,24]. Another study found recipients expressing vague expectations towards the offer but experiencing themselves being acknowledged through the dialogue [11]. The misunderstandings about the content and purpose of the dialogue as described by our participants can also shed light on the complexity inherent in the initiation of health-promoting dialogues with the target group. Other plausible explanations might be that older people might be more suspicious of offers from strangers due to earlier life experiences, or some might be affected by cognitive decline, making it difficult to interpret the situation and/or the offer.

Our informants described the recipients who accepted the offer as showing scepticism towards the offer before they knew and understood the dialogue’s content, a finding that is substantiated by a Swedish study [24]. Misunderstandings of the health-promoting purpose underscore the significance as well as the challenges of communicating health policy aimed at promoting measures among the target group at different levels of society. The described ambiguity of the dialogues’ purpose underlines the municipalities’ challenges in communicating the health promotion mission more clearly to older people. In addition, the lack of clarity is emphasised because the older population comprises a large and heterogeneous group with divergent needs and health challenges that appear at different stages of old age. From a health-promoting perspective, it is important to ask each individual what is meaningful in life to encounter their own motivation to preserve health. Such a conscious approach to marketing the offer should be essential regarding how to approach the target group. This attitude is also in line with the empowerment approach of participation and involvement [30,32] where the users are recognised as in charge of their own lives and are encouraged to make a difference using their resources [26].

Another reason for different perceptions among the target group regarding health-promoting dialogues can be linked to different perceptions of the concepts of ageing and old age, which might be the reason for declining the offered dialogue. One of our informants emphasised the importance of understanding old age through health-promoting dialogues. Behm et al. [46] found that older people either perceive the dialogue as too early in the ageing process or as irrelevant to their life situation, which has been confirmed in other studies [11,24]. Such perceptions might be interpreted as displacement mechanisms causing the non-uptake of important and relevant information and influences from healthcare professionals. Further, they could also illuminate how fear and uncertainty might be linked to negative thoughts about health, causing an unpreparedness for information dissemination concerning future health needs. Attitudes of older people towards the ageing process can influence their physical and psychological health, where positive attitudes are associated with making health-promoting choices despite reduced functional level or illness [47,48].

### 4.2. The Health-Promoting Mandate

Our informants conveyed the importance of observing the resources in each older person and building on these through dialogue about the person’s prerequisites for remaining independent in old age, which is reflected in policy recommendations on active ageing. Ambitions to promote active ageing through health-promoting dialogues are challenged when discussing the boundaries of a person’s influence over his own health, even if the individual has sufficient resources and skills to preserve personal health [2]. These arguments are contrasted by the heavy health-promoting emphasis on old age through the very concept of the World Health Organization [49] framework on active ageing, comprising the pillars of health, lifelong learning, participation, and security. In that regard, critical questions might be raised about the scope and relevance of such an “active” approach while meeting the individual older people in their homes. The arguments for the significance of active ageing have been used rhetorically in several national and international white papers through the decades, especially regarding necessary measures to meet the future pressure on the welfare state [50]. Such policy goals have been a counterweight to the traditional assumption of decline, degeneration, and withdrawal as normal features of ageing [51]. In this regard, the Norwegian researchers Blix and Ågotnes [50] question the rhetorical arguments based on a discourse analysis of some recent Norwegian public documents that govern both the content and goal of the Norwegian welfare state, for example, the report “A Full Life—All your Life—A Quality Reform for Older People” [5]. The discourse analysis brought forth three findings underpinning ideals of how older people should take responsibility for the goal of ageing well, designed as imperatives; Be physically active and healthy, be self-reliant, and be productive, resulting in an overall finding focusing on political and economic assumptions—Be a conscious consumer rather than a passive care recipient. Their findings could also be read between the lines when assessing the purpose of the health-promoting dialogues where our participants conveyed ideals of successful ageing as living independently and altering unhealthy habits to reach the goal of healthy ageing. The latter argument seems to add further complexity to the challenges in the complex conversation picture. Nevertheless, these arguments shed light on the challenging empowerment idealisation of individualism and the individual’s responsibility to optimise health and meaningfulness in the lives of older people. The different individual abilities need to be addressed, not only on a structural and political level but also on an individual level, such as by providing health-promoting dialogues.

### 4.3. Limitations of the Study

We believe that the study’s findings can enhance the knowledge base concerning health-promoting dialogues as an example of home-preventing visits to older people without municipal services. However, the study sample comprised three focus groups, with a total of nine participants representing only nurses and occupational therapists, producing limited interdisciplinary perspectives on the offer. In addition, the participants were interviewed digitally due to the pandemic prevailing during the data collection, which might have affected the group dynamics and the conversation flow. These issues might limit the transferability of the findings. However, the participants were recruited through a snowball recruitment process from some professionals’ wider networks, and digital interviews allowed the recruitment of participants from a wide geographical catchment area. We consider the broad recruitment area and the variety of experiences of each participant as the study finding’s strengthening elements. Additionally, when conducting digital meetings, larger groups might negatively affect conversation flow [52], indicating that smaller groups might facilitate dynamic, flexible, and spontaneous dialogues, which was experienced by the first author. The main analysis was carried out by a master’s student, but in close collaboration with experienced co-authors/supervisors. All authors contributed with critical reflections and discussions, producing a more nuanced final analysis that augmented the trustworthiness of the study.

## 5. Conclusions

This study revealed several complexities for the participants when facilitating health-promoting dialogue. Although these dialogues contributed to focusing on coping skills and health awareness among those receiving the offer, the participants needed to focus on ways to establish trustful relationships with older people. The dialogues also uncovered vulnerabilities, needs, and issues that older people were unaware of. The most complex aspect of the dialogues was the ambiguity caused by an unclear purpose of the offer towards the target group, as well as unclear expectations from potential recipients of the offer, which often resulted in a rejection of the dialogues. We discovered several reasons for the experienced ambiguity that are important to acknowledge to improve dialogue in service delivery. The dialogues should largely clarify their health-promoting purpose towards the target group, and avoid being associated with home care services, which might make older people unnecessarily sceptical. Further, the health-promoting mandate of such services might be interpreted as rhetorical in white papers on active ageing, arguing for a healthy and active population to take responsibility for their health during ageing. Rather than being supportive, such rhetoric might also be unnuanced, contributing to the empowerment idealisation of individualism when categorically optimising healthy and active ageing. The communication pattern of health-promoting dialogues with the public is of vital importance if one desires to succeed in establishing a dialogical relationship beyond initial contact.

In accordance with the measures of PHV [12,13], health-promoting dialogues could contribute to identifying coping skills, increasing awareness of recipients’ health, identifying health issues, and uncovering vulnerability. This offer serves as a link between a potential service user and information about healthcare services, which otherwise would be overlooked. The offer facilitates early intervention and prevention of health issues, thereby facilitating individuals’ ability to remain in their own homes for an extended period in line with the well-placed health policy on ageing. However, it is necessary to deeply investigate the exploitation of the potential inherent in these dialogues. In this regard, there is a need for more knowledge on how health-promoting dialogues could reach out to the target group and how the individual user could benefit the most from the offer. Another implication lies in reaching out to those who decline the offer for various reasons. A report on PHV in Norway provides some recommendations towards this challenge [13], including political and professional anchoring of the measure, awareness of aim, content, and target groups, recruitment of professionals with relevant expertise and personal suitability, as well as creating sufficient expectations among the target group through appropriate information in advance.

## Figures and Tables

**Figure 1 behavsci-14-00464-f001:**
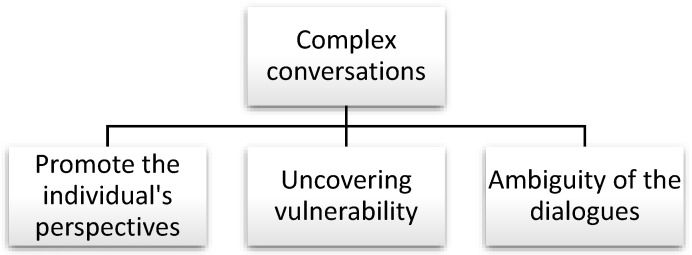
Overview of the theme and sub-themes.

## Data Availability

Data are contained within the article. Remaining data is unavailable due to ethical restrictions.

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
