# Peer review of "Health Professionals’ Experiences with Health-Promoting Dialogues for Older Home-Dwellers—A Qualitative Study"

_behavsci, 2024, doi:10.3390/bs14060464_

Round 1

Reviewer 1 Report (Previous Reviewer 2)

Comments and Suggestions for Authors

Thank you for the opportunity to review the manuscript again. Overall, a current topic for a broader readership and further exploration of this topic is certainly unique, especially to explore the experiences of healthcare professionals involved in health-promoting dialogues with home-dwellers aged over 75 years and without access to healthcare services.

A few questions / comments and suggestions:

In line 129-145, the limited information provided on the data analysis process. More details are needed on the coding process, development of themes, strategies used to ensure rigor, etc.

In line 425-434, it appears to have several limitations that should be addressed more fully in this limitation of the study section. The small sample size of 9 participants from 3 focus groups limits the transferability to the findings. The authors mention this briefly but do not sufficiently discuss how this impacts the results.

In this section, conducting the focus groups digitally due to COVID-19 could have impacted the group dynamics and conversation flow. This should also be discussed in more detail as a limitation.

In line 436-457, the conclusions section could provide more concrete practice implications from the study findings.

Comments on the Quality of English Language

There appear to be minor typos, grammatical errors, and awkward phrasing in some areas that need to be addressed through careful proofreading.

Author Response

Reviewer 2 Report (New Reviewer)

Comments and Suggestions for Authors

This is a very interesting study as it provides insight into the situation of the elderly population and the services offered to them from the perspective of the professionals working in these services.

Moreover, taking into account the increasing age of the population in practically all countries of the world, all studies that address the quality of life and the identification of our elderly are welcome and necessary. I am grateful for the opportunity to review the study.

In the introductory part, the authors have well described the situation and contextualization, and they have made a good bibliographic search.

Regarding the methodology, I think it is necessary to describe and add some data to better understand the study. Thus, in the method it would be interesting to add how long the interviews lasted, if each participant answered the questions one at a time, or if discussion was generated, if the interviews were recorded .....

Regarding the participants, I think it could be indicated how many years of experience they had (it is not enough to put more than two). I believe that professional experience is fundamental to be able to detect these needs and to be able to contribute to this type of study.

It would be interesting to know something more about the users attended by the professionals: if they attend more men or women (or the same), if they accept the service more in urban or rural areas, if there is a difference by level of education, marital status (with or without a partner, with or without children,...), if they live alone or not (if these data are available, of course).

The results are interesting, but I think it is necessary for the authors to incorporate in the discussion some ideas on how the service offered (or the dissemination of the same) could change taking into account the information obtained.

Since the increasing age of the population is an issue that occupies many countries worldwide, it would be important that the conclusions also include something regarding the generalizability of the results.

Finally, in the conclusions, more should be added about what implications; social, political or institutional; this work, or this type of work, has.

Author Response

Reviewer 3 Report (New Reviewer)

Comments and Suggestions for Authors

This paper describes a qualitative study examining healthcare professionals’ experiences involved in health-promoting dialogues with home-dwelling elders. The study has merit as it may help healthcare professionals and policy makers who want to improve the health of older adults.

I suggest conducting an extensive proof-reading of the entire article to improve its readability, as the phrasing of several sentences seemed cumbersome to me. The Introduction gives a good outline of the relevant background of the study, but I suggest expanding it with more thorough theoretical background. The Method section describes nicely the mothod, but I suggest to add a few more details about the methodology of the study. The results are described very well. The discussion addresses the main conclusions and limitations of the study and I suggest extending it just a bit.

In general, I think that this paper has important implications for researchers, policy makers and clinicians working with older individuals. Several modifications can improve the paper, as I suggest below. 

These are my specific comments to the authors: 

1.     Introduction: I suggest expanding the theoretical background to include other qualitative studies that described healthcare professionals experiences of working with elders in the context of health-promoting interventions.

2.     Introduction: I suggest expanding the paper’s theoretical framework. The authors describe empowerment as an important health-promoting tool. That is very good. I would expand that, such that the reader would be able to understand why health-promoting dialogues can be helpful for home-dwelling elders. 

3.     Sample and data collection (p. 3, second paragraph): I would refrain from using the term “represents” when describing the sample, as this sample is not a representative sample of the population, but rather a convenience sample.

4.     Analysis of data (p. 3, fifth paragraph): I would suggest explaining more about the data analysis: Were the focus groups’ meetings recorded and transcribed? How many researchers read the data and identified themes? Did they do it independently? Did they meet to converge and agree on the emerged themes?

5.     Ethical considerations (p. 4, first paragraph): Were the participants told that participation is voluntary and anonumous?

6.     Results: I would add a little more quotes for each subtheme, to help the reader understand the meaning of each subtheme. I suggest that the added quotes would describe the essence of the subtheme as it seemed to me that the given quoted did not catch the gist of the described subtheme.

7.     Discussion: It seems to me that the other possible reasons to the ambiguity of the aim of the health-promoting dialogues should be addressed. I guess that older individuals might be more suspicious of offers from strangers, maybe due to being afraid of elder-abuse or because of earlier life experiences of interpersonal difficulties. Cognitive decline might also be a possible reason for the difficulties many recipients experiences in understanding the aim of the dialogues. 

8.     Discussion: I think that this paper would benefit from specific recommendations for healthcare professionals conducting health-promoting dialogues: Maybe extending the meetings beyond one meeting, such that a better rapport would be possible and such that topics of vulnerability and further questions of the recipients may be addressed.

Comments on the Quality of English Language

Extensive proof-reading should be conducted on the entire paper.

Author Response

Reviewer 4 Report (New Reviewer)

Comments and Suggestions for Authors

Dear Authors,

I thoroughly went through your manuscript. In the following, please find my comments:

Title of article: Your investigation relates to four Norwegian municipalities. The title suggests a more extensive study. Please, revise.

Introduction: Please, add to this section the state of scientific knowledge on health-promoting dialogues in Norway (Dale/Westbye, Blix and Agotnes) and more detailed information on health-promoting dialogues as a public health offer (target groups, organisation, financing, frequency of visits).

Please, provide a more precise description of your theoretical/conceptual framework.

Methods: You mention that this manuscript bases on a previous master thesis. Please, insert source and moreover, indicate the originality of this article.

Provide more detailed information on the focus group interviews: Please, explain the necessity to compose three focus groups? Why is “homogeneity” of participants necessary? How long did the focus group interviews take? Who and how many persons hosted the focus groups?

Results: Insert source on page 4 (Kierkegaard).

Discussion: First paragraph is more appropriate for the conclusion section.

Limitation of the study: Statement (line 440 to 442) unclear.

Conclusion: Norwegian context missing; what about future research on the perspective of old(er) adults?

Kind regards!

Round 2

Reviewer 4 Report (New Reviewer)

Comments and Suggestions for Authors

Dear Authors,

In my opinion, the article is now suitable for publication. 

Kind regards!

This manuscript is a resubmission of an earlier submission. The following is a list of the peer review reports and author responses from that submission.

Round 1

Reviewer 1 Report

Comments and Suggestions for Authors

Thanks for the opportunity to review your paper.  Health promotion to the well-elderly is an increasingly important topic with the ageing of populations globally and the pressure this creates on health budgets.

Please find my comments below

Abstract

The abstract is adequate but could be improved with some small changes. I would like to know why these health professionals are approaching well, community-dwelling older people, for example is this part of a government/ local government program? How are people identified and chosen to be approached by health professionals? What particular health professionals? Are they primarily nurses, OTs, social workers etc.? This will help orient the reader and help interested readers to explore your article further.

Introduction

Overall the introduction is comprehensive, but I feel that if can be summarised. For example lines 49-53 and be shortened to tell the reader that in Norway there is increasing emphasis on PHV due to changes in policy or legislation and list the reports as references. What you need to present is the evidence for/against PHV as a programme. In the last paragraph you note that knowledge for executing PHV is scarce, but you have cited three reports that must have some evaluation of existing programmes?  

In general, an introduction presents what knowledge and evidence we have on a particular topic to date. For example, we should have information here about any research that has been undertaken about HVPs and health promotion dialogues and what the results have been and how, or if your approaches different from other countries. If there has been no research, you can simply state this. The last sentence – line 94 commences “based on existing knowledge…” but I am at a loss to understand what this knowledge is! (Remember you can draw on anecdotal reports if you justify these well).

The entire Introduction should build to be a clear justification for your study. You have written your aim clearly, (P.2, lines 94-96) but the rest of the introduction does not quite provide a logical argument why your focus groups with health professionals is worthwhile work and how this will add to knowledge of the topic.  

Please also note you have described healthcare professionals in the abstract and health personnel in the introduction. This means two different things to me, personnel could include non-professional care staff.

P2 line 52. I had not heard of Storting and had to look this up. You would be better to use a descriptor  e.g. “ and a report to the legislature of Norway “A full life….. etc”.  this will make you manuscript more accessible to an international audience.

Paragraph 2 on P.2 describes health dialogues targeting people who do not seek health services (presumably because they do not need them at the time) but leaves me wondering where and how these people are located? This material can be moved to the methods. Some further practical information such as how long do these dialogues go for? are they once only, or are their multiple visits? Are they linking with the older person GP? Etc.  

Materials and Methods

This section requires considerable work.  

The heading should be Methods. You do not describe any particular materials.

I disagree with the last sentence in 2.1 “In addition, the approach enables exploration and clarification of the informants’ perspectives, which cannot be achieved through individual interviews (26).” Please carefully justify this statement. Individual interviews most certainly allow and promote clarification of individual’s perspectives! I could guess you are describing the focus group process between participants, but you need to say what you mean.

Lines 114-115 “Participants were recruited through direct contact with four existing measures offered through four municipal services…” please state what these measures are.

Line 114 “Spring, 2021”. You are submitting to an international journal, and you need to ensure that country and even hemisphere specific references are thought through, and use wording is readily appreciable to an international audience.   

In the methods, details and consistency in terminology are critical. Eg Line 114 “interviews were conducted”.  You have stated you are conducting focus groups, so please keep your terminology consistent. Eg. line 116, 117 interviews and focus groups are both used.

A section on the nature of the questions you asked during focus groups is missing. This is a major omission. Please include at the very least a listing of key topic areas of the focus group.  A questioning guide as a table or box would be useful. I cannot interpret results without this information.

Reading though to the discussion, your summary in para 1 (p.6 lines 272-279)  would be better if you had presented some more detailed demographics about your focus group participants, including the length of time they had been in the HPV position and approximate numbers of older people they may have interacted with.  

It is usual to give details regarding ethics in this section, unless otherwise specified by the journal.

Results

As per above, I am not sure what your line of questioning was for your focus group participants, so I find I am ‘second-guessing’ your results.  The lack of information about the actual PHV program, the target group of older people and the nature of the ‘offer’, leaves me confused about what you are reporting in the results.

Your theme of ‘challenging dialogues’ seems to be more about challenges of clarity of the intention of the PHV programme and the rationale for the health professionals visit?

Sub theme 1. I feel lost! The aim is expressed as “health personnel’s’ experiences with health-promotion dialogues” but I am unsure if you are referring to the health professional or older person when presenting statements such as “mobilising one’s own power”.  Again at line 253 p.6 you state “There were many who declined – probably for various reasons, such as the experience of functional freshness,…”  seems to be reporting on the older people and not on the health professionals experience of reactions of decline of offer. Also, I do not understand what functional freshness is?  

Thinking though how you express your results will help with clarity.

Discussion

4.1 should be in the results  and not in the Discussion.

Again, the use of language to specify to whom you are referring is critical.  For example p. 6 lines 281 – 283 “Our informants emphasized a resource focus in meetings with the target group through the facilitated dialogue guide by highlighting a focus on the older person’s coping and awareness of their own health.”  You now have informants! Are these your focus group participants?  You have now mentioned a “facilitated dialogue guide”. Is this a manual for health professionals to use is the PHV program? If so it should have been explained and summarised in the introduction or methods.

Do not present new information in the discussion, unless you are specifically using it to explain, confirm or highlight dissent with your results. You have brought up a study “The significance of such a focus is confirmed in a Norwegian study on the target group’s experiences with health-promoting dialogue.” Why was this study not in the introduction? This may have provided important foundational information. You need to reference it too.

Similarly, P.7, line 326-327 “…a Norwegian study found that several older persons, who accepted the offered dialogue, also wanted follow-up consultations (16)” this sort of information needs to be reported in the Introduction to support how your study continues the work to understand this topic.  

You bring up some good points in the discussion concerning misunderstanding of dialogues and lack of cohesion between structural, political and ‘grass roots’ level of health professionals.

Minor comments

 It is good if you can have a native English speakers proof the document for spelling and expression. (Eg  watch for minor errors eg. P. 7 line 303 “a mutual and safe relation” should be ‘relationship’.) You can ask journal editors if they have an English language panel to assist with expression as well.  The manuscript could do with more concise expression.

Overall comments

Health promotion for older people is a growing area of interest and importance. I think you make some good points that are worthwhile publishing, but the paper is confusing to read, primarily due to lack of reporting key elements in each section. I strongly recommend that you read the American Psychological Association Publication Manual, particularly the section on what to include in each part of a manuscript.

Attention to consistency between the research question and how your results answer that question are critical – even if it is broad exploratory study. At this point I found the paper somewhat confusing to read, primarily due to omissions in introduction and the methods sections, as well as changing the labels given to ‘participants’.

Comments on the Quality of English Language

The English expression in this paper does require some work, but I have not addressed this as there are more pressing issues with major omissions in reporting.

Reviewer 2 Report

Comments and Suggestions for Authors

Thank you for the opportunity to review the manuscript. Overall, a current topic for a broader readership and further exploration of this topic is certainly unique, especially to investigate the antecedents of this intergenerational information sharing behavior from the youth to their elderly family members.

A few questions / comments and suggestions:

In line 94-96, it does not clearly state the aim or research questions of the study. These should be explicitly stated early in the introduction or methods section.

In line 98-120, ethical procedures around informed consent and ethics approval need to be described more clearly.

In line 139-270, the results section is overly descriptive and does not provide enough interpretation of the findings or link back to the aim and research questions. There should be more analytical depth in this section.

In line 271-415, the discussion section makes broad claims and generalizations beyond what the data seems to support. The limitations of the study should be acknowledged more clearly here.

In line 371, how to ‘linked to negative thoughts about health’, the relevant for the study is not clear.

In line 407, ‘moral obligation’ requires detail explanation, the relevant for the study is not clear.

Comments on the Quality of English Language

Overall, there are occasional grammar, style and formatting issues throughout that need to be corrected.